# Does Food Intake of Australian Toddlers 12–24 Months Align with Recommendations: Findings from the Australian Feeding Infants and Toddlers Study (OzFITS) 2021

**DOI:** 10.3390/nu14142890

**Published:** 2022-07-14

**Authors:** Najma A. Moumin, Merryn J. Netting, Rebecca K. Golley, Chelsea E. Mauch, Maria Makrides, Tim J. Green

**Affiliations:** 1Discipline of Pediatrics, Faculty of Health and Medical Sciences, University of Adelaide, Adelaide, SA 5000, Australia; najma.moumin@sahmri.com (N.A.M.); merryn.netting@sahmri.com (M.J.N.); maria.makrides@sahmri.com (M.M.); 2Women and Kids Theme, South Australian Health and Medical Research Institute, Adelaide, SA 5000, Australia; 3Nutrition Department, Women’s and Children’s Health Network, Adelaide, SA 5006, Australia; 4Caring Futures Institute, College of Nursing and Health Sciences, Flinders University, Adelaide, SA 5000, Australia; rebecca.golley@flinders.edu.au (R.K.G.); chelsea.mauch@flinders.edu.au (C.E.M.)

**Keywords:** toddlers, serving sizes, core foods, discretionary foods, Australia, dietary intake, survey

## Abstract

(1) Background: Food-based dietary guidelines promote population health and well-being through dietary patterns that reduce chronic disease risk while providing adequate energy and nutrients. In Australia, recommended dietary patterns based on servings per day from the five food groups—fruits, vegetables, cereals and grains, meats and alternatives, and dairy—have been developed for toddlers 1–2 years of age. However, no study has assessed the intake of the five food groups in this age group nationally. (2) Aim: To compare daily servings and the percentage of energy from the five food groups and discretionary foods in toddlers 1–2 years old to the Australian Dietary Guidelines. (3) Methods: Dietary intake was assessed using a one-day food record for 475 toddlers. (4) Results: Apart from fruit and dairy, servings of the five food groups were below the recommendations. Two-thirds of toddlers did not consume enough vegetables, and only 10% consumed the recommended number of servings for cereals and grains. On average, toddlers consumed only half the recommended servings of meat and alternatives. Nearly all toddlers (89%) consumed discretionary foods, which accounted for ~12% of total energy. Forty-five percent of toddlers received breastmilk. On average, breastfed toddlers consumed fewer servings from the five food groups than non-breastfed toddlers. Dairy contributed 20% of daily energy in all toddlers; however, this food group accounted for 13% in breastfed and 32% in non-breastfed toddlers on the day of the food record. (4) Conclusions: Compared to the recommendations, alignment with the servings of the five food group foods was not achieved by most toddlers, except for fruit and dairy. Discretionary foods may have displaced nutritious family foods. Consistent with Australian Infant Feeding Guidelines, many toddlers in this study continued to receive breastmilk but the recommended dietary patterns do not include breastmilk. Dietary modeling, including breastmilk as the primary milk source, is urgently needed, along with practical advice on incorporating breastmilk in a toddler’s diet while optimizing food consumption.

## 1. Introduction

Many countries have established food-based dietary guidelines to promote food intake that meets energy and nutrient requirements while minimizing the risk of obesity and chronic disease [1]. Dietary patterns have also been developed to guide food selection, including the suggested number of servings and serving sizes for each food group.

The Australian Dietary Guidelines were last updated in 2013 [2]. Along with the guidelines, a companion document [3] was developed to provide practical advice on meeting the guidelines. In Australia, there are five food groups, fruits, vegetables, cereals and grains, meats and alternatives, and dairy foods, along with an allowance for unsaturated fats and oils. In addition to the five food groups, the guidelines also limit the number of discretionary food servings—nutrient-poor and energy-dense foods and beverages higher in saturated fat, added sugars, salt, or alcohol [4]. Dietary modeling based on population intake data from nationally representative surveys was used to construct foundation diets and dietary patterns based on the recommended daily servings for each of the five food groups [5,6]. The serving sizes and the number of servings required from each food group were designed to meet the energy and nutrient needs for each sex and life stage group [7,8].

Due to a lack of Australian dietary intake data on children under 2 years, modeling for this age group was based on the dietary intake of 2–3-year-olds from the 1995 National Nutrition and Physical Activity Survey [5] and the 2007 Children’s National Nutrition and Physical Activity Survey [6,8]. Therefore, the dietary patterns for toddlers in the Australian Guide to Healthy Eating (AGHE) include the same five food groups and serving sizes as older age groups with smaller numbers of servings for each food group. However, discretionary foods are not recommended due to toddlers’ high nutrient needs relative to their low energy requirements [2].

In this paper, we compare the number of daily servings of the five food groups and discretionary foods consumed by toddlers enrolled in the Australian Feeding Infants and Toddlers Study (OzFITS) 2021 to dietary patterns recommended in the Australian Dietary Guidelines [2,9]. Due to the high breastfeeding rates in the second year of life [10], we compared the intake of the five food groups and discretionary foods in breastfed and non-breastfed toddlers. We also report the percentage of energy derived from each food group and the contribution of beverages, including breastmilk, to total energy intake.

## 2. Materials and Methods

OzFITS 2021 was a cross-sectional survey conducted with caregivers of children 0–24 months old [11,12]. Between April 2020 and April 2021, 1140 participants were enrolled across Australia, with recruitment proportional to population size in each state and territory. The Women’s and Children’s Health Network Human Research Ethics Committee (HREC/19/WCHN/44) approved the study, and all caregivers provided verbal consent.

Dietary intake data were collected using a one-day food record, with repeats on a non-consecutive day in 30% of the sample. This analysis includes dietary data collected from toddlers 12–24 months on the first day (*n* = 475). Caregivers were randomly allocated to a day of the week to record all foods and beverages consumed by their child. The details of the food record instrument [13,14,15] and follow-up interview [15] are described in detail elsewhere in the supplement [11]. All foods and beverages were entered directly into FoodWorks™ Professional Version 10 [16] using the 2011–2013 Australian Food, Supplement and Nutrient Database [17]. An additional database consisting of commercial infant and toddler foods (OzFITS Foods) was developed as these foods were largely unavailable in FoodWorks™.

All new foods, infant formula, and toddler milk entered in FoodWorks™ were assigned a unique 8-digit food group code based on the AUSNUT 2011-13 classification system [18]. Existing 8-digit codes in the AUSNUT 2011-13 Recipe database were applied to home-prepared mixed dishes with a similar composition. A new 8-digit code was created for unique recipes based on the major ingredient or the food group comprising the largest proportion of the dish [17,18]. Foods and recipes were further classified as discretionary or non-discretionary according to the criteria described in the Australian Bureau of Statistics discretionary flag list [4].

Breastmilk intake was estimated based on the number of minutes of active feeding to a maximum of ten minutes per feed, consistent with previous studies [19,20,21]. Breastfeeds lasting less than two minutes were excluded. Expressed breastmilk was entered as the reported quantity consumed by the child. The quantity (g) of dry formula or toddler milk consumed was calculated using the following equation: formula (g) = scoop weight (g)/prepared volume (mL) × consumed volume (mL). Similarly, the quantity of water consumed with formula was calculated as: water (mL) = water (mL)/total prepared volume (mL) × consumed (mL). The actual volume of water consumed was then added to the actual amount of formula or toddler milk consumed.

### Data Analysis

Dietary data were extracted from FoodWorks™ and linked with existing or unique 8-digit food group codes in the AUSNUT 2011-13 database [17] or OzFITS foods database. These codes permitted the identification and classification of foods into their respective food groups for analysis. Foods classified as five food groups included fruits, vegetables, cereals and grains, meats and alternatives, and dairy [2]. Other food groups were created for breast milk, formula and toddler milk, unsaturated fats and oils, and discretionary foods.

Serving sizes were assigned to five food group foods and unsaturated fats and oils correspond to those found in the Australian food selection guide [2]. Dairy servings with and without formula or toddler milks were reported. According to the manufacturer’s instructions, a serving of formula or toddler milk was defined as 250 mL and assumed to be prepared at full strength. Discretionary foods higher in fat, added sugars, and salt are not recommended for young children [2,9]. Since there is no allowance for discretionary foods in the recommended dietary patterns for children < 2 years, it is unclear if the serving size for discretionary foods (600 kJ) applies to this group [2]. For this reason, we applied a smaller serving size of 418 kJ/serving, consistent with toddler snacks. Foods and recipes classified as discretionary were excluded from the daily total for five food group servings. Daily servings of the five food groups and discretionary foods were calculated for the whole sample and then stratified according to breastmilk intake. Children were categorized as breastfed if they received breastmilk or were breastfed on the day of the food record. If they did not receive any breastmilk, they were categorized as not breastfed.

For mixed dishes, recipe data were disaggregated, and the gram weight of each food group was calculated using the following formula: portion (g) consumed/cooked recipe (g) × raw ingredient (g). Constituent parts were added to the daily total for each food group (g/day). The median (IQR) for each food group and beverage type was calculated for consumers only. Percent daily energy contribution from each food group and beverage type was calculated for both consumers and the whole sample. Total energy was attributed to the aggregated dish for mixed dishes based on the major ingredient or food group. For example, if a Bolognese pasta sauce recipe was 70% meat and 30% vegetable, the energy contribution was applied to the meat and alternatives food group.

## 3. Results

Demographic characteristics of caregivers and their toddlers who completed food records are described in Table 1. Approximately 80% of caregivers were university educated, and nearly all (97%) were the biological mothers of their children. One-half of caregivers were ≥35 years, and 40% had at least two children residing in the family home. The average age of children in the sample was 18 ± 3.4 months. Most children had a birth weight within the normal range of 2500–4499 g [22] and 73% (345/475) had commenced complementary feeding between their fifth and sixth months of life.

*Consumption of five food groups and discretionary foods:* The proportion of toddlers aged 12–24 months consuming five food groups and discretionary foods on the day of the food record and the median serving size for each group are presented in Table 2 and Table 3. Apart from unsaturated fats and oils, most children consumed foods from the five food groups on the day of the food record. Forty-two percent (97/233) of toddlers aged 12 to <18 months and 33% (80/242) of toddlers 18–24 months old did not consume any unsaturated fats or oils. Median daily servings for all five food group foods except for fruit and dairy were below the recommendations in both age groups. Over 90% of toddlers consumed meats and alternatives; however, this food group was consumed in small quantities, with both age groups consuming only half the recommended serving per day.

*Energy Intake:* The energy contribution from five food groups and discretionary foods are reported in Table 4 and Table 5. Nearly all children consumed cereals and dairy foods, and these foods contributed the most to daily energy intake. A total of 85–90% of toddlers consumed discretionary foods. Discretionary foods were the third-largest contributor to daily energy in older toddlers aged 18–24 months. In total, *n* = 1 toddler 12 to <18 months and *n* = 4 toddlers 18 to 24 months consumed only formula or toddler milk as their dairy source on the day of the food record.

*Main drinks consumed by toddlers on the day of the food record:* The types, amounts, and energy contribution from drinks are summarized in Table 6. One-half (51%) of toddlers 12 to <18 months and 40% of toddlers aged 18 to 24 months consumed breastmilk on the day of the food record, accounting for one-third and one-fifth of total energy, respectively. One-fifth of toddlers consumed formula or toddler milks, contributing 18–23% of total energy. Less than one-half of all toddlers consumed cow’s milk as the main drink. 

*Daily servings of five food groups and discretionary foods in breastfed and non-breastfed toddlers:* Consumption of five food groups and discretionary foods according to breastmilk intake on the day of the food record are presented in Appendix A. On average, non-breastfed toddlers consumed greater servings of all five food groups than breastfed toddlers. In non-breastfed toddlers, dairy foods contributed to 40% and 25% of daily energy for those 12 to <18 months and 18 to 24 months, respectively.

## 4. Discussion

Our study is the first to compare Australian toddlers’ intake of the five food groups and discretionary foods to the dietary patterns recommended in the Australian food selection guide. Most toddlers aged 12–24 months consumed foods from all five food groups; however, apart from fruit and dairy, the median servings of the five food groups were below the recommendations. Less than one-third of all toddlers consumed the recommended serving for meat and alternatives, and two-thirds did not consume the minimum recommended servings for vegetables. Only 10% of toddlers consumed the recommended servings of cereals and grains. In contrast, over half of toddlers consumed twice the amount of fruit recommended, and nearly half of toddlers consumed more than 1.5 servings of dairy foods. For the highest consumers (top quartile), dairy foods accounted for 30% of total energy. Despite no allowance for discretionary foods, 9 out of 10 toddlers consumed discretionary foods, contributing 8 to 15% of total energy for younger and older toddlers.

Our findings share some similarities with previous Australian studies of toddler dietary intakes from Adelaide and Brisbane (12–16 months) [20], Melbourne (18 months) [23], and Western Sydney (18 months) [24]. For example, the 20% energy coming from cereals and grains and 22% from dairy foods compare well with the 15–17% from cereals and grains and 28–35% of daily energy from dairy in the Adelaide/Brisbane and Western Sydney studies, respectively [20,24]. Consistent with these Australian studies [20,23,24], meat and meat alternatives in OzFITS 2021 were also consumed in small amounts with few toddlers consuming the recommended daily serving of 65 g. Factors contributing to low intakes from this food group may include the savory flavor profile [25] and the complex texture of these foods [26].

The National Health and Medical Research Council recommends cow’s milk and water as the ideal drinks for toddlers and limits sugar-sweetened drinks and fruit juice [9]. In addition, breastfeeding into the second year of life is encouraged [9]. Sugar-sweetened beverages were consumed by less than 5% of children, which is markedly lower than that reported in the 2016 US FITS at around 30% [27]. We also observed a lower percentage of toddlers (44%) consuming cow’s milk as a main drink compared to the US (83%). Over 50% of younger toddlers (12 to <18 months) and nearly 40% of older toddlers (18 to 24 months) in our survey consumed breastmilk as their main fluid on the day of food record. This is much higher than that reported in the 2010 Australian Infant Feeding Survey at 18% at 12 to <18 months and 7% of 18 to 24 months [28], and the US FITS 2016 at 18% and 5%, respectively [27]. Despite parents being advised to continue breastfeeding beyond 12 months [9], breastmilk is not included in the modeling used to inform the AGHE [8]. Given that around 45% of toddlers consumed breastmilk and breastmilk contributed to over 40% of energy for some toddlers, future modeling should include an allowance for breastmilk.

Due to the high breastfeeding rate in the second year of life [10], we stratified the consumption of core food servings and discretionary foods by breastmilk intake on the day of the food record to determine if any differences in consumption patterns for breastfed and non-breastfed toddlers existed. On average, breastfed toddlers consumed less of the five good group foods than non-breastfed toddlers, which may have implications for nutrient intake. Given that fortified cereals accounted for nearly half of all iron intake in toddlers in a Melbourne-based study, breastfed toddlers may be at greater risk of inadequate iron intake [19]. Similarly, toddlers in an Adelaide-based birth cohort study (mean 13.1 months) who received breastmilk as their sole milk source were at greater risk of inadequate intake of iron (50%) than those receiving formula (3%) with or without breastmilk [29]. A third of toddlers who received breastmilk only also had inadequate intakes of calcium compared to less than 10% for those who received other milks. Although breastmilk has the same amount of energy as toddler or cow’s milk, it contains less than a third of the calcium. Inadequate iron intakes were common in toddlers in our study and around 10% had low calcium intakes [12]. While the benefits of breastfeeding into the second year is beyond doubt, some toddlers may be receiving too much breastmilk as little practical guidance is given on how and when to offer breastmilk to optimize the consumption of core foods as toddlers transition from a milk-based diet to mostly family foods.

Our finding that discretionary foods are commonly consumed is consistent with other Australian studies [30,31,32]. For the highest quartile of the older toddlers, discretionary foods contributed up to one-quarter of daily energy intake, increasing the risk of obesity and displacement of other food groups in the diet. Commercial toddler foods marketed as healthy snacks in Australia [33] may be driving this as they were the most common foods consumed during snacking.

Our study has several strengths; chief among them is that it compared the five food group and discretionary food intake to the national recommendations. Furthermore, we have provided estimations of portion sizes consumed for each of the five food groups, which may be used to inform dietary modeling for this age group in subsequent revisions of the foundation diets [8]. In addition, we stratified the consumption of five food groups and discretionary food servings by breastmilk intake to illustrate differences in consumption patterns for breastfed and non-breastfed toddlers. As detailed in our previous paper, breastmilk intake may have been underestimated or overestimated as volume assumptions do not account for variation in feeding efficiency between toddlers [34]. However, the energy intakes for toddlers reported in our study [12] closely match the estimated energy requirements for this age group [7]. Maternal age is a strong predictor of continued breastfeeding [35] and may explain the high breastfeeding rates observed in our sample compared to previous Australian studies. Caregivers in our sample were highly educated and economically advantaged and may not be representative of the Australian population [11,36]. Finally, our analyses are based on a single day’s intake, which may not reflect usual intake; however, our findings are consistent with those reported in similar studies using 3-day weighed food records or 24 h recalls [23,24].

## 5. Conclusions

Toddlers enrolled in OzFITS 2021 consumed less than the recommended daily servings for all five food groups except for fruit and dairy. Of concern is that 9 out of 10 toddlers consumed discretionary foods, which may displace five food group foods in the diet. Milk consumption from all sources, including breastmilk, accounted for one-quarter of daily energy and may impact the nutritional adequacy of toddler diets. Given the high percentage of toddlers receiving breastmilk, there is an urgent need to develop dietary patterns that include breastmilk.

## Figures and Tables

**Table 1 nutrients-14-02890-t001:** Characteristics of caregivers and their toddlers, OzFITS 2021 (*n* = 475) ^1^.

	*n* (%)
**Caregiver Characteristics**	
Educational attainment (≥university degree)	364 (77)
Marital status (de facto or married)	446 (94)
Relationship to child (mother)	461 (97)
Age (years)	
<25	2 (0)
25–34	254 (54)
≥35	219 (46)
Country of birth	
Australia/New Zealand	351 (74)
Southeast Asia	31 (7)
United Kingdom/Ireland	30 (6)
Other	63 (13)
Number of children in the household	
1 child	291 (61)
2 children	139 (29)
≥3 children	45 (9)
Child characteristics	
Sex (female)	252 (47)
Age (months), mean ± SD	18 ± 3.4
Birthweight (g)	
<2500	24 (5)
2500–4499	444 (93)
≥4500	6 (1)
Solid food introduction (age, months)	
<4	6 (1)
4	100 (21)
5–6	345 (73)
≥7	24 (5)
Energy intake, kJ/day	
12 to <18 months	3902 (3359–4766)
18 to 24 months	4330 (3695–5125)

^1^ Data are presented as observed counts and percentages, mean ± standard deviation (SD), or median (IQR). OzFITS; Australian Feeding Infants and Toddler Study.

**Table 2 nutrients-14-02890-t002:** Daily servings of five food groups and discretionary foods consumed by toddlers aged 12–24 months, OzFITS 2021 (*n* = 475).

Food Group	Consumers *n* (%)	AGHE Serving Size ^1^	Recommended Servings per Day	Number of Servings per Day ^2^
Fruit	457 (96)	150 g	½	1.04 (0.51–1.62)
Vegetables	458 (96)	75 g	2–3	1.24 (0.56–2.31)
Cereals and grains	466 (98)	40 g bread	4	2.18 (1.28–3.18)
Meats and alternatives	432 (91)	65 g red meat	1	0.55 (0.23–1.10)
Dairy				
Without toddler milks	450 (95)	250 mL milk	1–1 ½	1.05 (0.51–1.71)
With toddler milks ^3^	455 (96)	250 mL milk	1–1 ½	1.33 (0.64–2.16)
Fats and oils	298 (63)	7–10 g	1	0.21 (0–0.70)
Discretionary foods	422 (89)	418 kJ	0	1.04 (0.36–2.27)

^1^ AGHE; Australian Guide to Healthy Eating; equivalent of 40 g bread, 65 g red meat, or 250 mL milk. For example, 40 g of cheese equals one serving of dairy [2]. ^2^ Values are median (IQR). The lower bound is used where a range exists for a recommended serving per day. ^3^ Dairy, including formula/toddler milk. A serving of formula/toddler milk is defined as 250 mL prepared volume.

**Table 3 nutrients-14-02890-t003:** Daily servings of five food groups and discretionary foods consumed by toddlers stratified by age, OzFITS 2021.

Food Group	Consumers *n* (%)	AGHE Serving Size ^1^	Recommended Servings per Day	Number of Servings per Day ^2^
**Toddlers 12 to <18 months (*n* = 233)**				
Fruit	224 (96)	150 g	½	0.98 (0.51–1.52)
Vegetables	229 (98)	75 g	2–3	1.26 (0.56–2.25)
Cereals and grains	225 (97)	40 g bread	4	1.99 (1.10–2.88)
Meats and alternatives	213 (91)	65 g red meat	1	0.49 (0.21–0.98)
Dairy				
Without toddler milks	220 (94)	250 mL milk	1–1 ½	1.04 (0.44–1.75)
With toddler milks ^3^	221 (95)	250 mL milk	1–1 ½	1.35 (0.57–2.24)
Fats and oils	136 (58)	7–10 g	1	0.12 (0–0.64)
Discretionary foods	201 (86)	418 kJ	0	0.78 (0.21–1.83)
**Toddlers 18 to 24 months (*n* = 242)**				
Fruit	233 (96)	150 g	½	1.12 (0.52–1.73)
Vegetables	229 (95)	75 g	2–3	1.19 (0.55–2.43)
Cereals and grains	241 (100)	40 g bread	4	2.39 (1.40–3.33)
Meats and alternatives	219 (91)	65 g red meat	1	0.60 (0.25–1.25)
Dairy				
Without toddler milks	230 (95)	250 mL milk	1–1 ½	1.11 (0.61–1.66)
With toddler milks ^3^	234 (97)	250 mL milk	1–1 ½	1.31 (0.71–2.04)
Fats and oils	162 (67)	7–10 g	1	0.29 (0–0.78)
Discretionary foods	221 (91)	418 kJ	0	1.47 (0.52–2.67)

^1^ AGHE; Australian Guide to Healthy Eating; equivalent of 40 g bread, 65 g red meat, or 250 mL milk. For example, 40 g of cheese equals one serving of dairy [2]. ^2^ Values are median (IQR). The lower bound is used where a range exists for a recommended serving per day. ^3^ Dairy, including formula/toddler milk. A serving of formula/toddler milk is defined as 250 mL prepared volume.

**Table 4 nutrients-14-02890-t004:** Energy and percentage total energy from five food groups and discretionary foods for toddlers aged 12–24 months, OzFITS 2021 (*n* = 475).

Food Group	Consumers *n* (%)	Energy, kJ/day ^1^	Percentage of Total Energy Intake ^1^
Fruit	457 (96)	367 (176–589)	9 (5–13)
Vegetables	458 (96)	173 (36–419)	4 (1–10)
Cereals and grains	466 (98)	859 (487–1269)	20 (12–30)
Meats and alternatives	432 (91)	297 (67–720)	9 (3–18)
Dairy			
Without toddler milks	450 (95)	709 (320–1146)	17 (8–27)
With toddler milks ^2^	455 (96)	888 (428–1495)	22 (10–34)
Fats and oils	298 (63)	53 (0–179)	1 (0–4)
Discretionary foods	422 (89)	435 (146–946)	10 (4–23)

^1^ Values are median (IQR). ^2^ Dairy, including formula or toddler milk. A serving of formula or toddler milk was defined as 250 mL prepared volume.

**Table 5 nutrients-14-02890-t005:** Energy and percentage total energy from five food groups and discretionary foods for toddlers stratified by age, OzFITS 2021.

Food Group	Consumers *n* (%)	Energy, kJ/day ^1^	Percentage of Total Energy Intake ^1^
**Toddlers 12 to <18 months (*n* = 233)**			
Fruit	224 (96)	362 (168–561)	9 (5–14)
Vegetables	229 (98)	173 (49–410)	4 (1–10)
Cereals and grains	225 (97)	773 (399–1137)	19 (11–29)
Meats and alternatives	213 (91)	297 (67–720)	8 (2–17)
Dairy			
Without toddler milks	220 (94)	691 (279–1135)	17 (7–28)
With toddler milks ^2^	221 (95)	853 (415–1509)	22 (10–37)
Fats and oils	136 (58)	31 (0–159)	1 (0–4)
Discretionary foods	201 (86)	305 (84–755)	8 (2–19)
**Toddlers 18 to 24 months (*n* = 242)**			
Fruit	233 (96)	372 (177–600)	9 (5–13)
Vegetables	229 (95)	173 (19–425)	4 (1–10)
Cereals and grains	241 (100)	931 (544–1421)	21 (13–31)
Meats and alternatives	219 (91)	428 (132–856)	11 (3–20)
Dairy			
Without toddler milks	230 (95)	733 (365–1136)	17 (9–27)
With toddler milks	234 (97)	890 (443–1469)	22 (11–32)
Fats and oils	162 (67)	74 (0–214)	2 (0–4)
Discretionary foods	221 (91)	617 (216–1116)	15 (5–25)

^1^ Values are median (IQR). ^2^ Dairy, including formula or toddler milk. A serving of formula or toddler milk was defined as 250 mL prepared volume.

**Table 6 nutrients-14-02890-t006:** The percentage of total energy from drinks for toddlers stratified by age, OzFITS 2021.

	Consumers *n* (%)	Intake, g/day ^1^	Energy, kJ/day ^1^	Percentage of Total Energy Intake ^1^
**Toddlers 12 to <18 months (*n* = 233)**				
Breastmilk	119 (51)	350 (200–540)	1000 (571–1542)	28 (13–42)
Formula/toddler milks	47 (20)	423 (227–619)	1057 (623–1614)	23 (13–37)
Cow’s milk (drink)	86 (37)	210 (116–411)	569 (312–1079)	15 (7–25)
Total milks ^2^	221 (95)	380 (206–541) ^2^	990 (570–1428)^2^	24 (14–38) ^2^
Water	228 (98)	243 (135–375)	−	−
Sweetened beverages	5 (2)	250 (156–231)	364 (251–445)	6 (5–9)
**Toddlers 18 to 24 months (*n* = 242)**				
Breastmilk	90 (37)	330 (200–513)	900 (564–1464)	22 (12–34)
Formula/toddler milks	42 (17)	354 (212–462)	869 (524–1252)	18 (10–30)
Cow’s milk (drink)	120 (50)	208 (129–324)	539 (315–822)	13 (8–19)
Total milks	211 (87)	300 (133–450)	792 (338–1232)	17 (8–30)
Water	230 (95)	289 (200–452)	−	−
Sweetened beverages	11 (5)	152 (104–263)	271 (163–473)	5 (3–8)

^1^ Values are median (IQR) for consumers of each beverage type except total milks, where values are for the whole sample. ^2^ Total milks are the sum of breastmilk, formula/toddler milks, cow’s milk, and alternatives consumed as a beverage.

## Data Availability

The data presented in this study are available on request from the corresponding author.

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
