# Peer review of "Does Food Intake of Australian Toddlers 12–24 Months Align with Recommendations: Findings from the Australian Feeding Infants and Toddlers Study (OzFITS) 2021"

_nutrients, 2022, doi:10.3390/nu14142890_

Round 1
Reviewer 1 Report
It is a well-written paper that addresses an understudied area. The topic/findings of this manuscript could be of interest to the readers of Nutrients. Overall, I don't have any major critiques, but I would like to request for the following changes/information:
1. This study recruited predominantly highly educated women and their toddlers. It is not a representation of the population of Australia. The authors need to mention this either in their title or introduction section that their sample population is from highly educated, and potentially mid-high income families. This also explains why the breastfeeding rate is so high in this study sample at 12-24 months of age.
2. That said, in their Discussion section they need to be careful to compare their findings with some other cohorts, especially some of the US national data, which tend to oversample low-income families.
3. Can the authors provide the total energy intakes of the toddlers in their sample by two age groups? I am very baffled by the serves recommended by the AGHE for each food group. If a toddler did meet all the recommended serves per day for each group, he/she might be consuming well over 2000 kcal, which is a lot of calories for a 12-24 month-old!
Author Response
Hello,
Thank you for your insightful comments. Please find attached our responses attached.
Najma

Reviewer 2 Report
Is sample size enough for Australian Toddlers?
L78 Dietary intake data were collected using a one-day food record. How representative is the sample?
L124 Children were categorized as breastfed if they received express breastmilk or were breastfed on the day of the food record. If they did not receive any breastmilk, they were categorised as not breastfed. Should not be limited to the day of record.
L158 Table 2 and 3 should be merged.
L178 Table 4 and 5 should be merged. How to evaluate total energy intake?
Author Response
Hello,
Thank you for your insightful comments. Please find our responses attached.
With thanks,
Najma

Round 2
Reviewer 2 Report
Breastfed or not should not be limited to the day of record.
Since the serves recommended by the AGHE is unreasonable, the evaluation is of little significance.